# Assessment of Starters of Lactic Acid Bacteria and Killer Yeasts: Selected Strains in Lab-Scale Fermentations of Table Olives (*Olea europaea* L.) cv. *Leccino*

Grazia Federica Bencresciuto [1], Claudio Mandalà [1], Carmela Anna Migliori [1], Giovanna Cortellino [2], Maristella Vanoli [2] and Laura Bardi [1,*]

[1] Research Centre for Engineering and Agro-Food Processing, CREA Council for Agricultural Research and Economics, 10135 Turin, Italy

[2] Research Centre for Engineering and Agro-Food Processing, CREA Council for Agricultural Research and Economics, 20133 Milan, Italy

* Correspondence: laura.bardi@crea.gov.it

**Abstract:** Olives debittering, organoleptic quality and safety can be improved with yeasts and lactic acid bacteria (LABs) selected strain starters, that allow for better fermentation control with respect to natural fermentation. Two selected killer yeasts (*Wickerhamomyces anomalus* and *Saccharomyces cerevisiae*) and *Lactobacillus plantarum* strains were tested for olive (cv. *Leccino*) fermentation to compare different starter combinations and strategies; the aim was to assess their potential in avoiding pretreatments and the use of excessive salt in the brines and preservatives. Lactobacilli, yeasts, molds, *Enterobacteriaceae* and total aerobic bacteria were detected, as well as pH, soluble sugars, alcohols, organic acids, phenolic compounds, and rheological properties of olives. Sugars were rapidly consumed in the brines and olives; the pH dropped quickly, then rose until neutrality after six months. The oleuropein final levels in olives were unaffected by the treatments. The use of starters did not improve the LABs' growth nor prevent the growth of *Enterobacteriaceae* and molds. The growth of undesirable microorganisms could have been induced by the availability of selective carbon source such as mannitol, whose concentration in olive trees rise under drought stress. The possible role of climate change on the quality and safety of fermented foods should be furtherly investigated. The improvement of olives' nutraceutical value can be induced by yeasts and LABs starters due to the higher production of hydroxytyrosol.

**Keywords:** table olives; fermentation; starter cultures; killer yeasts; lactic acid bacteria

## 1. Introduction

Table olives are a processed food widespread in the Mediterranean area obtained from fruits of *Olea europaea* L. Fruit processing is necessary to reduce their natural bitterness due to phenolic compounds, mainly oleuropein [1]. Phenolic compounds are among the most important components of olive drupes that contribute to taste and texture [2] and to their antioxidant, anti-inflammatory, antibacterial and cytotoxic properties [3]. Oleuropein has been extensively studied for its many beneficial properties due to antioxidant, scavenging of radicals, antimicrobial, antihypertensive and anticancer activities. During the ripening and fermentation of olives, oleuropein and ligstroside, another important phenolic compound, are hydrolyzed in a process involving the action of β-glucosidases that break the glycosidic bond with the formation of the aglycone forms of oleuropein and ligstroside, and esterases, which release elenoic acid, hydroxytyrosol (3,4 DHPEA) from oleuropein aglycone and tyrosol (p-HPEA) from ligstroside aglycone [4]. Tyrosol and hydroxytyrosol also act as scavengers of free radicals and as chelators of metals; furthermore, they have cardioprotective, anticancer and neuroprotective properties, and antimicrobial effects [5].

The abundance of phenolic compounds in table olives provides a high nutraceutical value to this fermented food.

Olive debittering can be reached by fermentation; the most commonly used processes are "Spanish style", in which green olives are soaked in diluted lye solution before fermentation in the brines, "Californian-style" in which olives are darkened by oxidation in lye solutions and air bubbling, and natural fermentation, also known as "Greek style", in which olives are immersed in 6–10% NaCl brines and fermentation takes place lasting 8–10 months due to the growth of autochthonous microorganisms of olives and plants [6–9]. Yeasts and lactic acid bacteria (LABs) are the main agents of natural fermentations due to their ability to grow in high NaCl concentrations: they rapidly acidify the brines so that high salinity and low pH prevent the growth of spoilage or pathogenic microorganisms. Moreover, LABs and yeasts can significantly improve the sensory properties of processed olives producing secondary metabolites [9–11]. LABs have been widely assessed and featured for their ability to improve the olives quality and safety through brine acidification and release of bacteriocines, whereas yeasts can play a double role, as they may also be agents of the deterioration of the fruits, gas-pocket formation, softening of the olive flesh, production of off-flavors and odors [12], or cause brines clouding or package swelling [13].

Furthermore, yeasts can contribute to debittering, promote LAB growth, improve flavor, taste and texture of final product, inhibit the growth of spoilage yeasts and molds by producing killer toxin and inhibit several adverse microorganisms. Due to these positive actions, lesser use of salt and additives could be allowed [9,13–16]. Moreover, yeasts can make up for LABs when their activity is inhibited by high phenols and/or salt concentrations or excessively low pH [17].

The industrial production of black and green olives by spontaneous fermentation is subject to autochthonous microflora, physicochemical conditions, olive varieties, fermentable substrates and brine salt content; the results can be unpredictable [18]. The growth of undesirable strains can lead to abnormal fermentation and a non-compliant final product [6]. Better control of the fermentation can be reached with a starter culture of selected LABs and yeast strains. Selected strains are exploited to inhibit spoilage due to undesirable microorganisms or pathogens, reduce debittering times, and obtain a better final product. Starters could also be useful for producing probiotic table olives [19].

Among the species of LABs, *Lactobacillus plantarum* and *Lactobacillus pentosus* strains have been mainly characterized, while the yeasts most used as starters are *Wickerhamomyces anomalus*, *Saccharomyces cerevisiae* and *Candida boidinii* [20].

In recent years, several studies focused on killer yeasts as they can inhibit the growth of spoilage yeasts by producing killer toxins in fermented foods, including olives [9,14–16]. Among them, *W. anomalus* and *S. cerevisiae* killer strains were selected for olive fermentation [14]. The mechanisms of action can be variable and influenced by various factors; a killer toxin can inhibit DNA replication, induce membrane permeability changes and arrest the cell cycle [21–23].

In the present work, the use of two selected yeast strains for olives fermentation was studied to evaluate their potential to obtain a good final product that can avoid the pretreatments of olives, excessive salt in the brines and further additives and preservatives. *W. anomalous* and *S. cerevisiae* strains previously characterized for killer activity and debittering capability were chosen. Olives cv. *Leccino* were fermented in the brines with the inocula of starter cultures of these yeast strains, alone or associated with a selected strain of *L. plantarum*, in comparison to spontaneous fermentation.

## 2. Materials and Methods

### 2.1. Olive Samples

Natural black olives cv. *Leccino* were provided by Romeo Ficacci s.r.l., a table olive industry located in Castelmadama (Rome, Italy). Black olives (70 kg) were collected at the BBCH (Biologische Bundesanstalt, Bundessortenamt and Chemical Industry) phenological

stage 8, maturity of fruit [24] and washed with tap water to eliminate plant materials (residues of leaves, branches) and superficial contaminants.

### 2.2. Microbial Strains and Starter Production

Two selected oleuropeinolytic killer yeast strains (collection of the University of Sassari, Italy), *Wickerhamomyces anomalus* (Wa1) and *Saccharomyces cerevisiae* (Sc24), and one selected bacterial strain, *Lactobacillus plantarum* (B51) (collection CREA-IT, Pescara, Italy) were used as starters.

Bacterial strain precultures were obtained in 100 mL MRS (Man Rogosa and Sharpe: Peptone 10 g/L, Beef Extract 10 g/L, Yeast Extract 5 g/L, Glucose 20 g/L, Di-potassium Hydrogen Phosphate 2 g/L, Sodium Acetate 5 g/L, Di-amonium Citrate 2 g/L, Magnesium Sulphate 0.2 g/L, Manganous Sulphate 0.05 g/L and Tween® 80 1 g/L) broth incubated at 30 °C for 24 h. Bacterial cells were then adapted to the saline environment of the brine by incubation in 100 mL MRS broth supplemented with 3% NaCl at 30 °C for 24 h, followed a further incubation in 100 mL MRS broth supplemented with 6% NaCl at 30 °C for 24 h.

*S. cerevisiae* and *W. anomalus* precultures were produced in 200 mL of YEPD (10 g/L Yeast Extract, 20 g/L Peptone, 20 g/L Dextrose) broth at 28 °C for 48 h. Afterward, the yeasts were adapted to the salinity of the brines by sequential incubation at 28 °C for 48 h in YEPD broth supplemented with 3% NaCl, then in YEPD broth with 6% NaCl.

Yeast and bacteria cells were collected by centrifugation (3500 rpm at 15 °C for 15 min), then inoculated in 8% NaCl sterile brine to achieve an initial population of $10^6$ CFU/mL in each jar according to the experimental setup.

### 2.3. Experimental Set Up

Lab-scale fermentations were carried out in 5 L capacity glass jars according to the Greek-style method. 2.5 kg of olives were placed in each jar, then filled with 2 L of 8% NaCl (*w/v*) previously sterilized at 121 °C × 20 min.

Seven different fermentation conditions were compared: (i) spontaneous fermentation (SP), (ii) fermentation inoculated with *L. plantarum* B51 (LP), (iii) fermentation inoculated with *W. anomalus* (WA), (iv) fermentation inoculated with mixed *L. plantarum* and *S. cerevisiae* (MIX1), (v) fermentation with sequential inoculation of *L. plantarum* followed one month later by *S. cerevisiae* (LY1), (vi) fermentation inoculated with mixed *L. plantarum* and *W. anomalus* (MIX2), (vii) fermentation with sequential inoculation of *L. plantarum* followed one month later by *W. anomalus* (LY2).

All treatments were performed in triplicate (three fermentation jars per treatment) at room temperature (between 19 °C and 25 °C) for 188 days. Brine samples (10 mL) and olives (20 drupes) were collected aseptically and subjected to microbiological and chemical analysis throughout the fermentations.

### 2.4. Microbiological Analysis

Brines were analyzed to determine the content of Lactobacilli, Yeasts, Molds, *Enterobacteriaceae* and Total Aerobic Bacteria.

The samples (1 mL) were aseptically transferred to 9 mL of sterile saline solution. Serial dilutions in the same sterile saline were prepared, and 1 mL or 0.1 mL samples of the appropriate dilutions were spread or mixed on the following media: Man Rogosa and Sharpe Agar plates for the detection of Lactobacilli, 3M™ Petrifilm™ Aerobic Count Plates for the detection of Total Aerobic Bacteria, 3M™ Petrifilm™ Enterobacteriaceae Count Plates for the detection of *Enterobacteriaceae*, and 3M™ Petrifilm™ Yeast and Mold Count Plates for the detection of yeast and mold.

The 3M™ Petrifilm™ Plate is a sample-ready culture medium system that contains nutrients, a cold-water-soluble gelling agent, and an indicator system that facilitates microorganism enumeration. The 3M™ Petrifilm™ formulation are (i) Violet Red Bile with Glucose nutrients, cold-water soluble gel, and tetrazolium indicator (3M™ Petrifilm™ Enterobacteriaceae Count Plates, (6420/6421), Manufactured at Brookings, SD, USA (ISO

9001:2015, FM 14552)), (ii) Standard Methods nutrients, cold-water soluble gel, and tetrazolium indicator (3M™ Petrifilm™ Aerobic Count Plates, (6400/6406/6442), Manufactured at Brookings, SD, USA (ISO 9001:2015, FM 14552)), (iii) Nutrients supplemented with antibiotics, a cold-water-soluble gel, and indicator (3M™ Petrifilm™ Yeast and Mold Count Plates, (6407/6417/6445), Manufactured at Brookings, SD, USA (ISO 9001:2015, FM 14552)).

Incubation was carried out at 28 °C for 5 days for Lactobacilli, at 37 °C for 24 h for *Enterobacteriaceae*, at 28 °C for 5 days for yeasts and mold and at 37 °C for 48 h for Total Aerobic Bacteria. Results were expressed as colony-forming units per milliliters (CFU/mL) of brines.

Fungal isolation was performed directly from emerging colonies that developed on 3M™ Petrifilm™ plates by detaching them diligently with a sterilized bacterial loop and transferring them on PDA (4 g/L Potato extract, 20 g/L Dextrose, 15 g/L Agar) plates. Monoconidial cultures were produced for each isolate to obtain pure fungal colonies. Each monoconidial culture was incubated at 25 °C for 48 h; at the end of the incubation period, colony characteristics (color, mycelium, colony type and shape) were observed. Dimensions of available conidia and hyphal features (color, shape, presence or absence of chlamydospores) were recorded.

*2.5. Physico-Chemical Analysis*

2.5.1. Sugars and Alcohols

Sugars and alcohols were extracted from olives following the method described by Fibiani et al. [25]. Brine samples and extract from olives pulp were analyzed by HPLC after filtration through 0.45 μm Nylon filters. Sucrose, glucose, fructose, mannitol, sorbitol, ethanol and glycerin were analyzed by chromatographic analysis with HPLC (Jasco PU-980, Jasco, Tokyo, Japan) equipped with a RI530 refractive index detector, combined with a Chromnav data acquisition system version 3.32, in a Benson 87C carbohydrate column (300 mm × 8 mm) maintained at 80 °C. The mobile phase was ultrapure water (HPLC grade), 0.7 mL/min flow. Commercial reference standards were used to create calibration curves [25].

2.5.2. Organic Acids

Brine samples were analyzed after filtration through 0.45 μm Nylon filters. An Agilent 1200 Series HPLC was used for organic acid analysis coupled to a UV/Vis detector with detection at 214 nm. The column used was a Repromer H + column (300 mm × 8 mm), maintained at 63 °C. The mobile phase was $H_2SO_4$ 25 nM, 0.5 mL/min flow. Commercial reference standards of citric, malic, pyruvic, succinic, lactic and acetic acids were used to create calibration curves [25].

2.5.3. pH

All pH measurements were made in a Crison BASIC 20 + pH-meter (Crison Instruments S.A., Alella, Spain) fitted with a Crison electrode. The calibration was made with standard buffers at pH 4.00 and 7.00. The determinations were executed in triplicate.

2.5.4. Phenolic Compounds

The brines were filtered through 0.45 μm Nylon filters and diluted 1:1 in a methanol/acidified water mixture (80/20). Extraction from destoned olives was carried out following the method of Ambra et al. [2], except for the use of frozen samples homogenized by an Ultraturrax IKA T18 Basic (IKA Works Inc., Wilmington, NC, USA) for 30 s in the presence of the extraction mixture and treatment of the homogenized sample in an ultrasonic bath (Elmasonic S, Singen, Germany) for 30 min instead of liquid nitrogen treatment. The samples were then filtered with 0.45 μm membranes to remove any residue. Subsequently, olives extracts and brine samples were analyzed by HPLC technique following the method described in Bleve et al. [26], using a Jasco BS-997 HPLC equipped with a Supelco ODS80Ts RPC18 column (250 mm × 4.6 mm, 5 μm). All analyses were performed in triplicate. Phenol

compounds were quantified using calibration curves of authentic phenolic standards, with concentrations between 250 ppm and 1000 ppm for oleuropein, between 125 ppm and 750 ppm for tyrosol and between 250 ppm and 1000 ppm for hydroxytyrosol.

*2.6. Rheological Properties of Olive*

The rheological properties of olives were measured by texture profile analysis (TPA) and a puncture test. Both analyses were performed by means of a TA.XT Plus texture analyzer (Stable Micro System Ltd., Godalmig, UK) and data acquisition and integration by using Texture Exponent 32-bit software. All analyses were done at room temperature. For both tests, twenty olives were analyzed for each jar and sixty for each treatment. The texture was evaluated only at the end of treatment. Each olive was centered horizontally under the probe for measurement.

2.6.1. Texture Profile Analysis (TPA)

The olives were subjected to a double compression cycle using a cylindrical flat probe (10 cm diameter) at a test speed of 1 mm/s and a deformation of 15% of the initial size. By the acquired force (g)/displacement (mm) curve, multiple textural parameters were acquired:

- Hardness: Maximum force of the first compression.
- Springiness: Ratio between the distance traveled during the first compression cycle and the distance during the second cycle. It is the rate at which a deformed sample returns to its original size.
- Cohesiveness: Ratio between the area (work) during the second compression and the area (work) during the first compression. It is the degree to which a product can be deformed before it breaks.
- Gumminess: Hardness × cohesiveness. It is the energy required to disintegrate the product to the state ready for swallowing.
- Chewiness: Hardness × cohesiveness × springiness. It is the number of chews needed to masticate the product until it is ready for swallowing.

2.6.2. Puncture Test

A puncture test was carried out using a needle probe (2 mm diameter) with a crosshead speed of 6.67 mm/s. From the force curve, the following parameters were extracted: firmness corresponding to the maximum force (firmness, g), rigidity index corresponding to the slope of the last part of the force curve (dF/dS, g/mm) and work required to puncture the olive (g × mm).

*2.7. Chemicals*

Reference standards of HPLC-grade oleuropein, tyrosol and hydroxytyrosol were purchased from Cayman Chemicals (Ann Arbor, MI, USA). References standards (HPLC grades) of sugars, alcohols and organic acids; HPLC-grade solvents (methanol, acetonitrile, hexane) and acetic acid 99–100% were purchased from Sigma-Aldrich (Milan, Italy). HPLC-grade Ultrapure water was purchased from Carlo Erba reagents (Milan, Italy), standard buffers at pH 4.00 and 7.00 from CRISON (Crison Instruments S.A., Alella, Spain). MRS Broth with tween 80, BIOLIFE, Milan, Italy; Sodium Chloride, SACCO s.r.l., Cadorago (Como), Italy; YEPD, Sigma-Aldrich, Milan, Italy; MRSCM0361, Oxoid, Hampshire, United Kingdom; 3M™ Petrifilm™ count plates specific for the detection of yeast, mold, *Enterobacteriaceae* and total aerobic bacteria were purchased from 3M Italy, Pioltello, Milan (Italy).

*2.8. Statistical Analyses*

The data were subjected to analysis of variance (ANOVA) and post-hoc LSD test, using $p \leq 0.05$ as the cut-off level of significance. ORIGIN software (OriginPro, Version 2022. OriginLab Corporation, Northampton, MA, USA) was used to check for significant differences among treatments at each sampling time.

## 3. Results

### 3.1. Microbial Analysis

Among the checked microbial populations (Figure 1), Lactobacilli showed the fastest and highest growth at the early phase of fermentation; then, their cell concentrations showed a decreasing trend (Figure 1A). On the 40th day, they reached cell concentrations between $3.6 \times 10^7$ CFU/mL in LY1 and $9.4 \times 10^7$ CFU/mL in MIX2 treatments; moreover, cell concentrations were highest where *W. anomalus* selected strain was inoculated in the mix with *L. plantarum* (40th day) or alone (63rd day), while they were lowest when *L. plantarum* was associated to *S. cerevisiae*. The total aerobic bacteria grown at 37 °C showed a trend similar to LABs, but at slightly lower concentrations (Figure 1C); during the third month of fermentation, they were significantly lower in treatments inoculated with both yeasts and LAB selected strains. *Enterobacteriaceae* followed an inverse trend throughout the fermentation: their cell concentrations rose from the 77th to the 140th day, then dropped sharply on the 188th day (Figure 1D). On the 40th day, yeast growth (Figure 1B) was highest in the SP treatment; later, the trend changed, showing that, in general, cell concentration rose until the 77th day, followed by a subsequent descent.

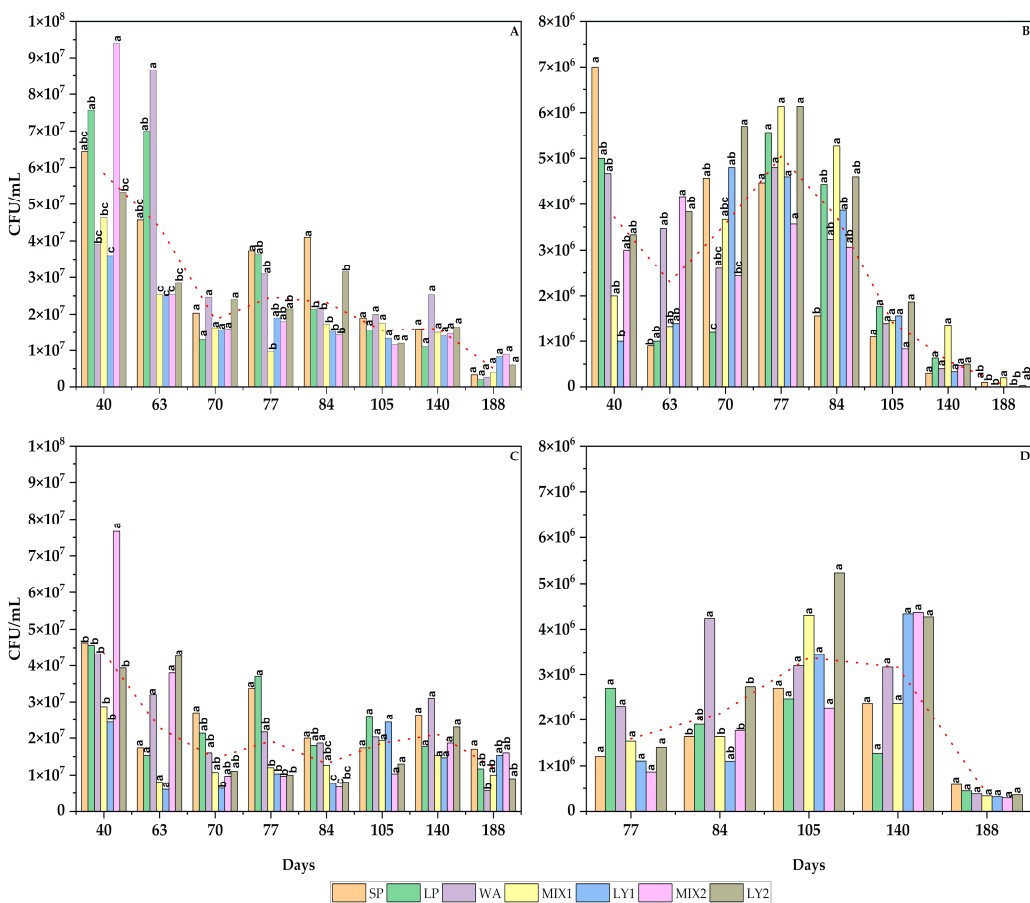

**Figure 1.** Lactobacilli (**A**), yeasts (**B**), total aerobic bacteria (**C**) and *Enterobacteriaceae* (**D**) in the brines of the different treatments. The data are expressed as means of the triplicate measurements followed by ANOVA tests. Significant differences are indicated by different letters ($p \leq 0.05$). The red dotted line represents the average microbial counts for each sampling time.

Maximum yeast cell concentration was reached on the 70th day in SP and LY, on the 77th day in LP, WA, MIX1, MIX2 and LY2 treatments; it sharply dropped on the 84th day in SP and on the 105th day in the other treatments. From the 63rd day onward, the presence of molds was observed, reaching significant abundance during the late phase of fermentation. High variability detected in terms of quantification did not allow significant differences

to be evidenced among treatments, with the exception of the last sampling time (188th), where the molds were lowest in MIX1, which was significantly lower than in LP treatment. From monoconidial cultures obtained from isolated colonies, it was possible to identify the *Fusarium* spp. genera as the prevalent mold in all treatments except for the MIX1 treatment. In particular, the microscopic observations of the morphological characteristics allowed the identification of *Fusarium solani* as the prevalent species. Specifically, the *F. solani* traits are reported in the Supplementary Materials (Figure S1).

### 3.2. Physico-Chemical Analysis

### 3.2.1. Sugars and Alcohols

The compositions of the sugars and alcohols of the raw olive pulp are reported in Table 1. The most abundant sugar was glucose (9477.74 ppm ± 203.12) followed by sucrose (703.93 ppm ± 125.43) and fructose (615.80 ppm ± 26.43). Among sugar alcohols, mannitol showed the highest concentration (3619.97 ppm ± 570.04), while the sorbitol concentration was 92.41 ppm ± 66.39, with the latter showing very high variability (CV = 72%).

**Table 1.** Sugar, alcohol and polyphenol content in the raw olive pulp. Data are the average from the three replicates ± standard error.

| Sugars and Alcohols | ppm |
|---|---|
| Sucrose | 703.93 ± 125.43 |
| Glucose | 9477.74 ± 203.12 |
| Fructose | 615.80 ± 26.43 |
| Mannitol | 3619.97 ± 570.04 |
| Sorbitol | 92.41 ± 66.39 |
| **Phenolic Compounds** | **ppm** |
| Oleuropein | 682.09 ± 230.58 |
| Hydroxytyrosol | 1632.23 ± 675.70 |
| Tyrosol | 1816.50 ± 545.35 |

During fermentation, the olives' sugar content rapidly decreased, with glucose being the most rapidly consumed (Table 2). At the 23rd day, about 76% of the glucose was consumed on average, followed by sucrose (45%) and fructose (38%), whereas among sugar alcohols (Table 3), sorbitol strongly decreased (62%), while the decrease in mannitol was only 8%. Glucose and sucrose were more rapidly consumed in SP, but fructose was more rapidly consumed in LY2. On the 40th day of fermentation, glucose and sucrose were almost completely consumed (96% and 94% respectively on average), while 26% fructose residue was still present, with the lowest concentrations found in MIX1, LY1, MIX 2 and LY2; also, mannitol showed a strong decrease from 23rd to 40th day of fermentation, when its concentration dropped to 15% mean residue, highest in LP, lowest in SP. All sugars and sugar alcohols were almost zeroed on the 105th day in all treatments; only traces of fructose were found in treatments LP, MIX2 and LY2, and glucose in treatment SP. At the same sampling time, a high ethanol concentration was detected in olives, ranging from 468 ppm to 852 ppm in SP and WA treatments, respectively. Sucrose, after zeroing on the 105th day, was detected at 73 ppm mean concentration on the 140th day and 30 ppm on the 188th day. Sorbitol and glycerol concentrations showed the same trend during fermentations in olive pulp, rising throughout the fermentations to the highest values on the 140th day (mean values 424 ppm and 1450 ppm, respectively) (Table 3).

**Table 2.** Change in concentrations of sugars in the pulp of *Leccino* olives during spontaneous (SP) or starter-driven fermentation, measured by HPLC analysis. Different letters indicate significant differences among the different treatments ($p \leq 0.05$) at the same sampling time. Data are averaged from the three replicates $\pm$ standard error.

| Day | 23 | 40 | 105 | 140 | 188 |
|---|---|---|---|---|---|
| **Treatments** | | | Sucrose (ppm) | | |
| SP | 305.49 ± 4.85 [e] | 52.76 ± 3.88 [b] | 2.16 ± 0.25 [a] | 80.71 ± 1.73 [a] | 0.00 ± 0.00 [d] |
| LP | 426.14 ± 6.52 [ab] | 124.19 ± 7.67 [a] | 0.00 ± 0.00 [b] | 103.55 ± 6.64 [a] | 0.00 ± 0.00 [d] |
| WA | 410.69 ± 14.47 [abc] | 141.17 ± 17.57 [a] | 0.00 ± 0.00 [b] | 94.39 ± 3.51 [a] | 11.78 ± 8.98 [cd] |
| MIX1 | 386.85 ± 19.64 [c] | 0.00 ± 0.00 [c] | 0.00 ± 0.00 [b] | 93.32 ± 3.03 [a] | 133.98 ± 12.17 [a] |
| LY1 | 397.47 ± 5.29 [bc] | 0.00 ± 0.00 [c] | 0.00 ± 0.00 [b] | 31.69 ± 2.33 [c] | 40.14 ± 2.31 [b] |
| MIX2 | 340.79 ± 7.95 [d] | 0.00 ± 0.00 [c] | 0.00 ± 0.00 [b] | 72.33 ± 6.96 [ab] | 23.25 ± 8.05 [bc] |
| LY2 | 439.57 ± 11.71 [a] | 0.00 ± 0.00 [c] | 0.00 ± 0.00 [b] | 34.10 ± 9.14 [bc] | 0.00 ± 0.00 [d] |
| | | | Glucose (ppm) | | |
| SP | 1336.53 ± 30.22 [f] | 264.96 ± 5.08 [c] | 5.14 ± 4.06 [ab] | 225.35 ± 64.80 [a] | 0.00 ± 0.00 [b] |
| LP | 2678.28 ± 89.35 [ab] | 589.65 ± 138.75 [b] | 0.73 ± 0.73 [b] | 20.60 ± 1.27 [b] | 0.00 ± 0.00 [b] |
| WA | 2370.15 ± 74.86 [cd] | 886.36 ± 209.44 [a] | 0.00 ± 0.00 [b] | 18.90 ± 3.25 [b] | 17.17 ± 7.52 [a] |
| MIX1 | 2091.05 ± 104.84 [e] | 231.74 ± 17.91 [c] | 0.00 ± 0.00 [b] | 12.64 ± 0.94 [b] | 9.27 ± 2.12 [ab] |
| LY1 | 2730.10 ± 38.10 [a] | 172.82 ± 7.96 [c] | 0.00 ± 0.00 [b] | 4.93 ± 0.72 [b] | 3.80 ± 0.55 [b] |
| MIX2 | 2472. 06 ± 34.42 [bc] | 217.64 ± 18.48 [c] | 1.38 ± 0.20 [b] | 0.00 ± 0.00 [b] | 0.00 ± 0.00 [b] |
| LY2 | 2204.15 ± 108.27 [de] | 212.25 ± 3.30 [c] | 10.17 ± 4.36 [a] | 2.54 ± 0.37 [b] | 0.00 ± 0.00 [b] |
| | | | Fructose (ppm) | | |
| SP | 425.31 ± 11.99 [ab] | 234.09 ± 3.06 [a] | 1.05 ± 0.87 [b] | 37.69 ± 14.64 [a] | 0.00 ± 0.00 [c] |
| LP | 366. 63 ± 33.94 [ab] | 377.97 ± 52.56 [a] | 5.62 ± 0.87 [b] | 1.92 ± 0.56 [b] | 0.00 ± 0.00 [c] |
| WA | 379.57 ± 13.07 [ab] | 248.65 ± 114.49 [a] | 0.00 ± 0.00 [b] | 0.00 ± 0.00 [b] | 17.52 ± 2.73 [b] |
| MIX1 | 468.63 ± 59.45 [a] | 60.56 ± 7.77 [b] | 0.00 ± 0.00 [b] | 3.29 ± 0.73 [b] | 44.27 ± 3.28 [a] |
| LY1 | 374.99 ± 7.21 [ab] | 83.75 ± 9.93 [b] | 0.00 ± 0.00 [b] | 2.91 ± 0.69 [b] | 6.24 ± 2.15 [c] |
| MIX2 | 359.85 ± 12.16 [ab] | 42.41 ± 20.16 [b] | 21.93 ± 6.53 [b] | 12.01 ± 4.14 [b] | 0.00 ± 0.00 [c] |
| LY2 | 287.43 ± 118.51 [b] | 53.84 ± 8.28 [b] | 63.09 ± 19.98 [a] | 20.09 ± 10.27 [ab] | 0.00 ± 0.00 [c] |

**Table 3.** Change in concentration of alcohols in the pulp of *Leccino* olives during spontaneous (SP) or starter-driven fermentation, measured by HPLC analysis. Different letters indicate significant differences among the different treatments ($p \leq 0.05$) at the same sampling time. Data are average from three replicates $\pm$ standard error.

| Day | 23 | 40 | 105 | 140 | 188 |
|---|---|---|---|---|---|
| **Treatments** | | | Mannitol (ppm) | | |
| SP | 2841.82 ± 184.23 [bc] | 330. 47 ± 15.47 [b] | 1.03 ± 0.52 [a] | 107.09 ± 13.69 [b] | 66.47 ± 20.02 [b] |
| LP | 2721.22 ± 443.95 [c] | 881.01 ± 406.13 [a] | 0.00 ± 0.00 [b] | 0.00 ± 0.00 [d] | 666.53 ± 20.87 [a] |
| WA | 4248.15 ± 117.43 [a] | 533.65 ± 12.23 [ab] | 0.00 ± 0.00 [b] | 0.00 ± 0.00 [d] | 608.07 ± 111.80 [a] |
| MIX1 | 4205.16 ± 192.19 [a] | 613.97 ± 89.47 [ab] | 0.00 ± 0.00 [b] | 99.60 ± 10.39 [b] | 123.15 ± 16.39 [b] |
| LY1 | 3505.10 ± 184.46 [ab] | 460.93 ± 31.92 [ab] | 0.00 ± 0.00 [b] | 66.89 ± 5.74 [c] | 103.76 ± 19.62 [b] |
| MIX2 | 3146.29 ± 29.75 [bc] | 445.93 ± 51.91 [ab] | 0.00 ± 0.00 [b] | 166.02 ± 6.45 [a] | 0.00 ± 0.00 [b] |
| LY2 | 2539.49 ± 361.20 [c] | 466.11 ± 72.69 [ab] | 0.00 ± 0.00 [b] | 183.03 ± 9.68 [a] | 43.10 ± 7.92 [b] |

**Table 3.** *Cont.*

| | Day | 23 | 40 | 105 | 140 | 188 |
|---|---|---|---|---|---|---|
| | | | | **Sorbitol (ppm)** | | |
| SP | | 37.29 ± 13.06 [ab] | 125.09 ± 3.96 [bcd] | 0.13 ± 0.08 [a] | 808.75 ± 39.18 [a] | 90.20 ± 9.68 [c] |
| LP | | 27.99 ± 5.01 [ab] | 97.59 ± 0.44 [cd] | 0.00 ± 0.00 [b] | 321.37 ± 41.25 [bc] | 46.20 ± 5.41 [d] |
| WA | | 36.42 ± 5.99 [ab] | 59.67 ± 27.49 [d] | 0.00 ± 0.00 [b] | 402.61 ± 54.24 [bc] | 113.06 ± 0.20 [b] |
| MIX1 | | 66.90 ± 16.76 [a] | 211.87 ± 39.55 [b] | 0.00 ± 0.00 [b] | 235.62 ± 75.17 [c] | 198.39 ± 8.32 [a] |
| LY1 | | 18.58 ± 7.61 [b] | 201.85 ± 10.38 [bc] | 0.00 ± 0.00 [b] | 432.52 ± 46.01 [b] | 0.00 ± 0.00 [e] |
| MIX2 | | 16.25 ± 2.69 [b] | 337.44 ± 84.51 [a] | 0.00 ± 0.00 [b] | 290.22 ± 88.32 [bc] | 16.42 ± 4.77 [e] |
| LY2 | | 41.03 ± 26.42 [ab] | 177.08 ± 8.37 [bc] | 0.00 ± 0.00 [b] | 479.53 ± 86.09 [b] | 10.22 ± 4.86 [e] |
| | | | | **Ethanol (ppm)** | | |
| SP | | 0.00 ± 0.00 | 0.00 ± 0.00 | 467.75 ± 132.03 [d] | 0.00 ± 0.00 | 0.00 ± 0.00 [b] |
| LP | | 0.00 ± 0.00 | 0.00 ± 0.00 | 822.96 ± 32.33 [ab] | 0.00 ± 0.00 | 0.00 ± 0.00 [b] |
| WA | | 0.00 ± 0.00 | 0.00 ± 0.00 | 851.71 ± 78.11 [a] | 0.00 ± 0.00 | 0.00 ± 0.00 [b] |
| MIX1 | | 0.00 ± 0.00 | 0.00 ± 0.00 | 598.26 ± 50.76 [bcd] | 0.00 ± 0.00 | 0.00 ± 0.00 [b] |
| LY1 | | 0.00 ± 0.00 | 0.00 ± 0.00 | 773.19 ± 40.86 [abc] | 0.00 ± 0.00 | 0.00 ± 0.00 [b] |
| MIX2 | | 0.00 ± 0.00 | 0.00 ± 0.00 | 576.26 ± 23.88 [cd] | 0.00 ± 0.00 | 41.49 ± 5.33 [a] |
| LY2 | | 0.00 ± 0.00 | 0.00 ± 0.00 | 535.13 ± 102.12 [d] | 0.00 ± 0.00 | 0.00 ± 0.00 [b] |
| | | | | **Glycerol (ppm)** | | |
| SP | | 0.00 ± 0.00 | 0.00 ± 0.00 [c] | 0.00 ± 0.00 | 1342.62 ± 287.81 [a] | 2362.26 ± 496.59 [a] |
| LP | | 0.00 ± 0.00 | 0.00 ± 0.00 [c] | 0.00 ± 0.00 | 1331.19 ± 52.87 [a] | 1487.78 ± 150.80 [bc] |
| WA | | 0.00 ± 0.00 | 470.96 ± 156.54 [b] | 0.00 ± 0.00 | 1353.50 ± 156.54 [a] | 950.54 ± 137.67 [cd] |
| MIX1 | | 0.00 ± 0.00 | 740.64 ± 228.03 [b] | 0.00 ± 0.00 | 1726.35 ± 49.04 [a] | 1792.01 ± 77.12 [ab] |
| LY1 | | 0.00 ± 0.00 | 880.76 ± 74.01 [ab] | 0.00 ± 0.00 | 1681.73 ± 207.55 [a] | 450.85 ± 32.96 [de] |
| MIX2 | | 0.00 ± 0.00 | 1221.38 ± 274.93 [a] | 0.00 ± 0.00 | 1345.29 ± 137.99 [a] | 0.00 ± 0.00 [e] |
| LY2 | | 0.00 ± 0.00 | 858.80 ± 73.42 [ab] | 0.00 ± 0.00 | 1368.59 ± 205.04 [a] | 0.00 ± 0.00 [e] |

In the brines, only traces of sugars were found at early sampling times and throughout the fermentations (Figure 2A,B). Sucrose was not detected. The highest concentrations at the last sampling time were 278.40 ppm glucose (treatment MIX2) and 187.20 ppm fructose (treatment LY2). Instead, sugar alcohols were detected at higher concentrations (Figure 2C,D): mean values of mannitol were 2405.53 ppm on the 16th day and 1521.34 ppm on the 105th day, and sorbitol appeared on the 188th day at an average concentration of 594.94 ppm. Ethanol concentration rose, reaching mean concentrations of 45,873.69 ppm on the 16th and 3822.06 ppm on the 140th day, then a more rapid increase was observed until the 180,581.91 ppm mean concentration on the 188th day (Figure 3D). Glycerol was not detected in the brines.

### 3.2.2. Organic Acids

Lactic acid, acetic acid and pyruvic acid reached the highest concentration in the brines very rapidly (Figure 3A–C). On the 16th day, lactic acid, acetic acid, and pyruvic acid were highest in WA (4100.25 ppm, 32,819.01 ppm and 956.21 ppm, respectively) and lowest in MIX2 (1207.10 ppm, 5875.03 ppm and 394.55 ppm, respectively). Then, a decrease was observed in pyruvic acid in all treatments, lactic acid in all treatments except MIX2, and acetic acid in all treatments except MIX2 and LY2.

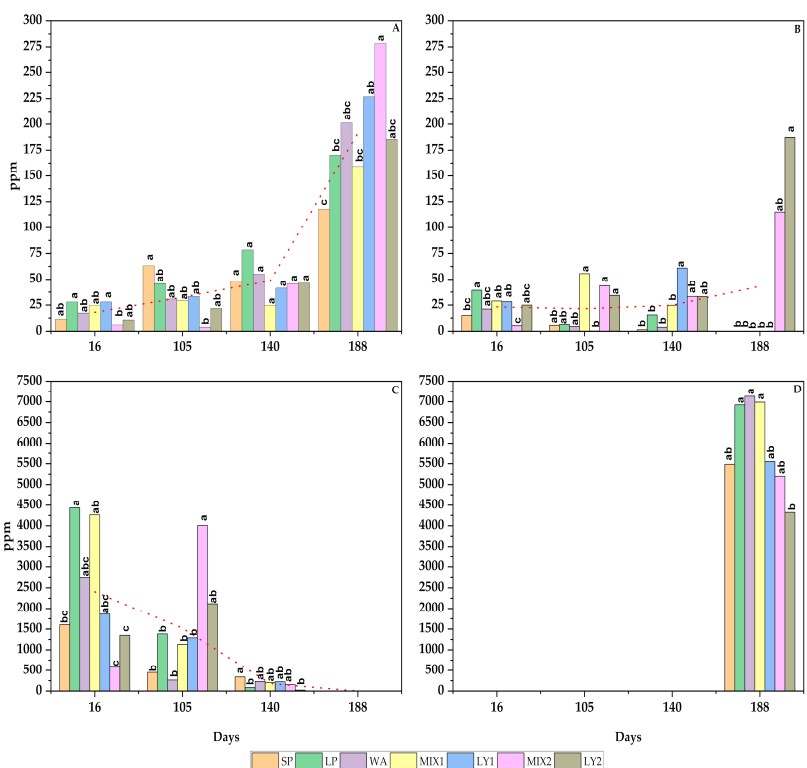

**Figure 2.** Glucose (**A**), fructose (**B**), mannitol (**C**) and sorbitol (**D**) evolution of the brines during fermentation. The data are expressed as means of triplicate measurements. Significant differences (LSD) are indicated by different letters ($p \leq 0.05$). The red dotted line represents the average values at each sampling time.

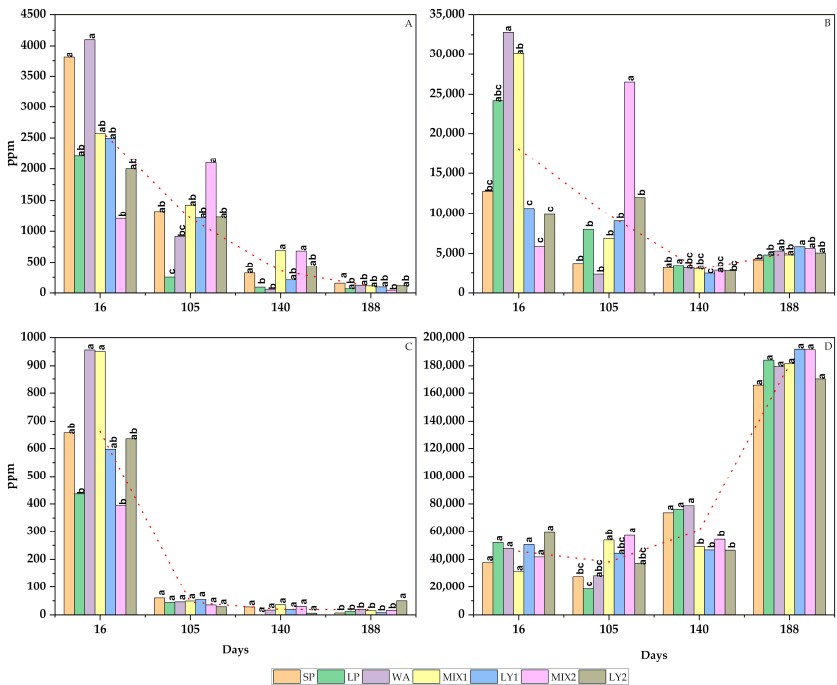

**Figure 3.** Lactic acid (**A**), acetic acid (**B**) and pyruvic acid (**C**) and ethanol (**D**) values in the brines during fermentations. The data are expressed as averages of triplicate measurements. Significant differences (LSD) are indicated by different letters ($p \leq 0.05$). The red dotted line represents the average values at each sampling time.

Citric acid concentrations were very low in the brines throughout the fermentations, but an increase was observed at the last sampling time on the 188th day when it reached a mean concentration of 294.17 ppm (Figure 4A). Malic acid was detected in the brines already on the 16th day at concentrations between 1075.4 ppm in WA and 415.07 ppm in MIX2, then it decreased to almost disappearing on the 140th day; from the 16th to 105th day, it strongly decreased in all treatments except for the MIX2 (Figure 4B). The succinic acid concentration in the brines was very low on the 16th day of fermentation, then an increase was observed. On the 105th and 140th days, the mean concentrations were 214.72 ppm and 199.03 ppm, respectively, while they further decreased on the 188th day except for in the LY2 treatment (Figure 4C).

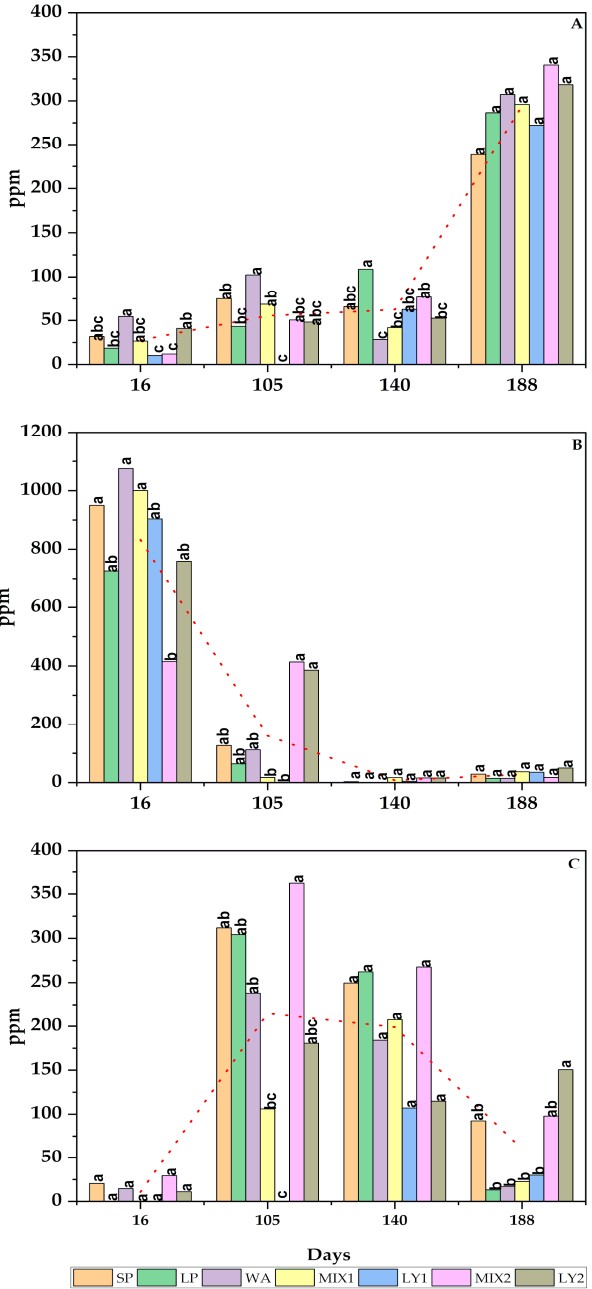

**Figure 4.** Citric acid (**A**), malic acid (**B**) and succinic acid (**C**) in the brines. The data are expressed as averages of the triplicate measurements. Significant differences (LSD) are indicated by different letters ($p \leq 0.05$). Red dotted line represents the average values at each sampling time.

### 3.2.3. pH

pH levels quickly dropped to values between 4.37 (LY1) and 4.43 (SP) in the second week of fermentation (Figure 5). A slow increase was observed starting from the 40th day, reaching a 5.02 mean value on the 70th day and almost neutrality (6.76 mean value) at the end of fermentation. Significant differences were found between the treatments from the start, with lower pH in the inoculated treatments compared to spontaneous fermentation; these differences became gradually more pronounced, evidencing lower pH in treatments with the inocula of both yeasts and LAB strains (MIX1, LY1, MIX2, LY2), with the lowest values detected in the MIX2 and LY2 treatments at the last sampling time.

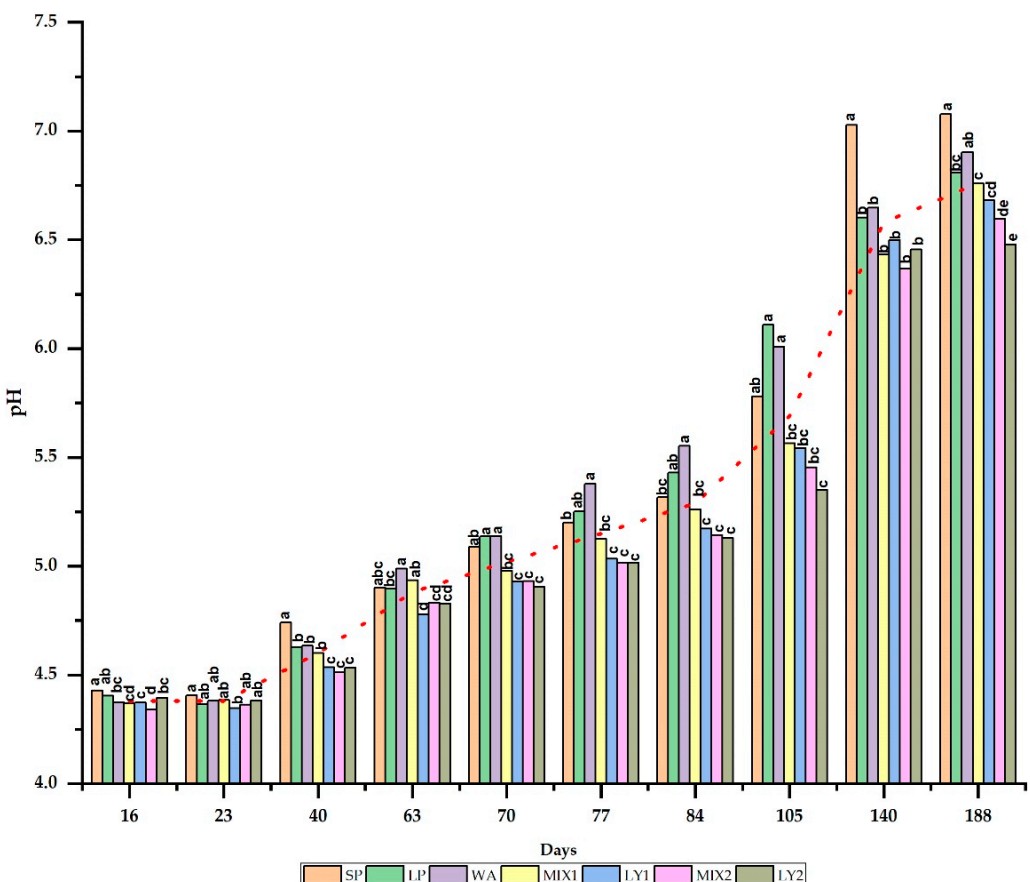

**Figure 5.** Brine pH values during fermentation. The data are expressed as averages of the triplicate measurements. Significant differences (LSD) are indicated by different letters ($p \leq 0.05$). The red dotted line represents the average pH values at each sampling time.

### 3.2.4. Phenolic Compounds

Oleuropein aglycone, hydroxytyrosol and tyrosol were quantified in raw fruits (Table 1) and during fermentation both in the brines (Figure 6A–C) and olive pulp (Table 4) during fermentations.

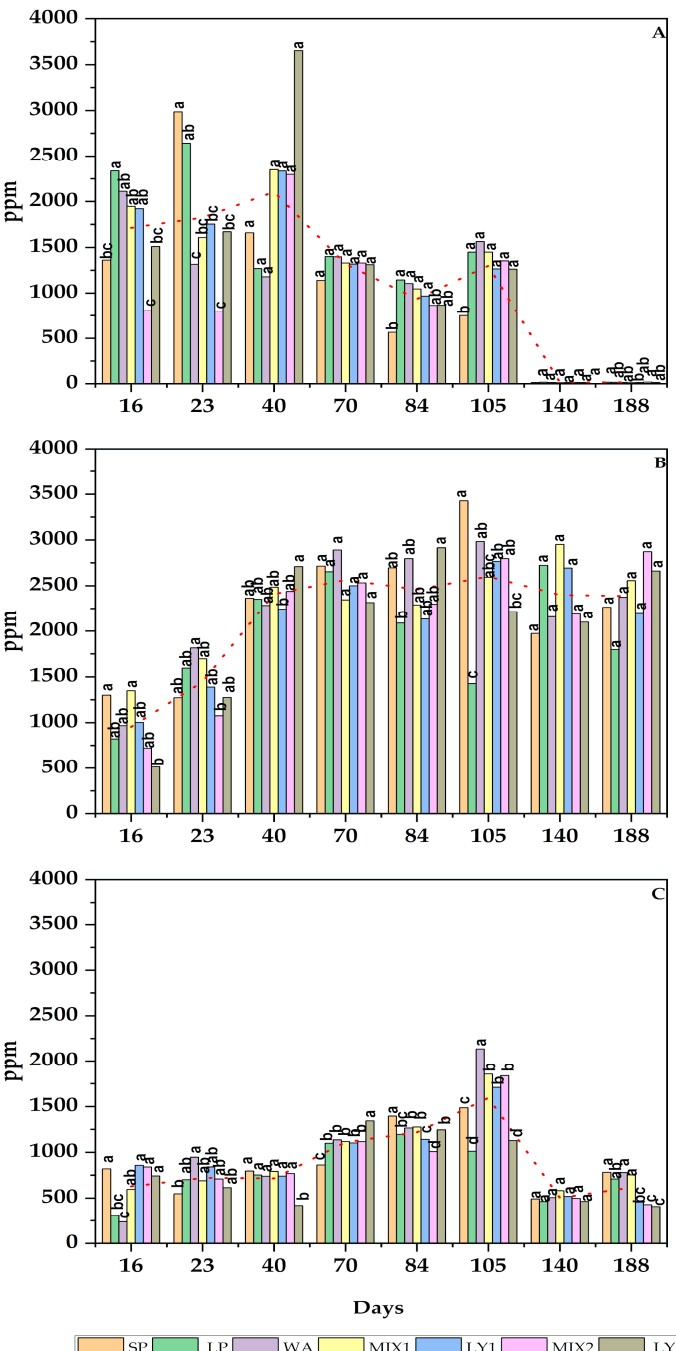

**Figure 6.** Polyphenol compound concentrations, oleuropein aglycone (**A**), hydroxytyrosol (**B**) and tyrosol (**C**) in the brines during fermentation. The data are expressed as averages of the triplicate measurements. Significant differences (LSD) are indicated by different letters ($p \leq 0.05$). The red dotted line represents the average of a single molecule for each sampling time.

**Table 4.** Change in concentration of oleuropein, hydroxytyrosol and tyrosol in the pulp of olives during fermentations, measured by HPLC analysis. Different letters indicate significant differences (LSD) among the different treatments ($p \leq 0.05$) within the same sampling time. Data are the averages from three replicates $\pm$ standard error.

| Day | 23 | 40 | 105 | 140 | 188 |
|---|---|---|---|---|---|
| **Treatments** | | | **Oleuropein (ppm)** | | |
| SP | 2368.83 ± 1194.84 [bc] | 555.58 ± 502.88 [a] | 258.69 ± 26.71 [a] | 724.3 ± 680.23 [a] | 435.61 ± 63.41 [a] |
| LP | 4709.37 ± 455.77 [a] | 342.6 ± 37.11 [a] | 228.39 ± 51.85 [a] | 313.01 ± 198.86 [a] | 456.20 ± 159.71 [a] |
| WA | 4281.64 ± 647.67 [a] | 1887.59 ± 2613.48 [a] | 242.86 ± 53.21 [a] | 421.81 ± 144.07 [a] | 328.18 ± 53.63 [a] |
| MIX1 | 3821.5 ± 159.55 [ab] | 398.43 ± 146.34 [a] | 379.31 ± 173.46 [a] | 410.80 ± 108.86 [a] | 545.42 ± 368.49 [a] |
| LY1 | 3969.62 ± 1211.10 [a] | 374.93 ± 32.43 [a] | 283.02 ± 97.63 [a] | 415.54 ± 81.80 [a] | 337.83 ± 193.36 [a] |
| MIX2 | 2470.98 ± 1182.53 [bc] | 358.59 ± 30.75 [a] | 270.93 ± 49.17 [a] | 450.19 ± 117.43 [a] | 318.80 ± 98.89 [a] |
| LY2 | 1367.09 ± 394.85 [c] | 384.06 ± 61.36 [a] | 230.36 ± 63.59 [a] | 466.44 ± 77.55 [a] | 290.03 ± 21.01 [a] |
| | | | **Hydroxytyrosol (ppm)** | | |
| SP | 2009.22 ± 1752.86 [a] | 2130.88 ± 823.28 [b] | 2428.56 ± 174.15 [a] | 1940.845 ± 860.32 [a] | 1917.57 ± 879.92 [ab] |
| LP | 2881.68 ± 78.045 [a] | 3567.21 ± 257.33 [a] | 2381.63 ± 399.71 [a] | 1675.35 ± 397.48 [a] | 1725.02 ± 865.91 [ab] |
| WA | 3574.43 ± 251.95 [a] | 4034.87 ± 90.911 [a] | 2494.15 ± 357.76 [a] | 1506.91 ± 995.07 [a] | 1877.16 ± 184.35 [ab] |
| MIX1 | 1991.57 ± 1059.90 [a] | 3878.65 ± 398.48 [a] | 2379.66 ± 932.88 [a] | 2143.85 ± 315.19 [a] | 1015.06 ± 105.44 [b] |
| LY1 | 2292.77 ± 852.84 [a] | 3531.42 ± 544.91 [a] | 1768.14 ± 1134.18 [ab] | 1813.44 ± 582.48 [a] | 1258.71 ± 666.4 [ab] |
| MIX2 | 3397.36 ± 479.51 [a] | 3741.07 ± 214.48 [a] | 1131.05 ± 1794.00 [b] | 2407.07 ± 1195.88 [a] | 1275.51 ± 239.39 [ab] |
| LY2 | 2822.89 ± 828.89 [a] | 3033.28 ± 1070.70 [ab] | 1411.71 ± 897.04 [b] | 2276.69 ± 573.87 [a] | 2254.53 ± 439.46 [a] |
| | | | **Tyrosol (ppm)** | | |
| SP | 719.41 ± 221.50 [b] | 379.89 ± 260.59 [a] | 163.25 ± 6.44 [ab] | 413.38 ± 103.69 [a] | 277.7 ± 40.36 [a] |
| LP | 767.18 ± 55.87 [b] | 192.77 ± 9.35 [b] | 150.4 ± 13.28 [b] | 357.33 ± 21.34 [a] | 304.11 ± 32.64 [a] |
| WA | 1069.78 ± 170.11 [ab] | 219.16 ± 5.20 [ab] | 161.03 ± 29.77 [ab] | 365.74 ± 80.85 [a] | 255.81 ± 24.90 [a] |
| MIX1 | 913.75 ± 65.43 [ab] | 211.62 ± 13.81 [ab] | 219.47 ± 56.96 [a] | 359.77 ± 36.99 [a] | 310.21 ± 75.51 [a] |
| LY1 | 978.33 ± 345.91 [ab] | 205.96 ± 20.49 [ab] | 190.66 ± 58.56 [ab] | 398.12 ± 37.70 [a] | 244.64 ± 75.15 [a] |
| MIX2 | 1187.25 ± 379.71 [a] | 216.53 ± 21.95 [ab] | 188.65 ± 25.85 [ab] | 391.93 ± 71.35 [a] | 284.12 ± 55.74 [a] |
| LY2 | 759.83 ± 99.78 [b] | 200.88 ± 21.34 [b] | 192.52 ± 36.58 [ab] | 323.54 ± 84.63 [a] | 261.95 ± 35.40 [a] |

Oleuropein in the brines (Figure 6A) was detected starting from the first sampling time (16th day), with starting values within a range of 800 ppm and 2300 ppm, with significant differences among treatments. The highest value was recorded in LP (2336.57 ppm). Oleuropein values gradually decreased during the six months of fermentations, with significant differences among treatments in almost all sampling times. The highest percentage of oleuropein depletion (between 98.26% and 99.14%) was recorded for all treatments between the 105th and 140th day after the beginning of the test.

After 188 days, all the samples reached contents of oleuropein aglycone ranging between 12 ppm and 18 ppm, with slight but significant differences among treatments. MIX1 showed the lowest content (12.52 ppm), while SP had the highest level of oleuropein in the brine at the end of the process, with a value similar to MIX2.

In contrast with the trend of oleuropein, hydroxytyrosol in the brines (Figure 6B) gradually increased in almost all the treatments up to the 105th day. At early fermentation, after 16 days, there was a great variability of the concentration of this molecule among treatments, with values ranging between 513.00 ppm (LY2) and 1349.04 ppm (MIX1). Between 105 and 140 days, a decrease was noticed in SP, WA and MIX2 theses, while for LP and MIX1, a significant increase was recorded. At the end of fermentation, the values were all similar (between 1800 ppm and 2800 ppm) for all treatments without significant differences.

Following the same trend of hydroxytyrosol, tyrosol (Figure 6C) in the brines showed a gradual growth up to 105th days, where the highest value was showed by WA treatment (2134.88 ppm) followed by a decrease (54–69%) on the 140th days. Between 140 and 188 days, SP, LP, WA and MIX1 increased their content in tyrosol, while in the other theses, the values were quite constant. During the process, SP was the treatment with lower concentrations of tyrosol except for the sampling on the 16th, 40th and 84th days. Instead, at the end of fermentation, SP, WA, LP and MIX1 showed higher values ranging

between 708.80 ppm (LP) and 781.64 ppm (SP), while the other treatments ranged between 406.93 ppm (LY2) and 457.40 ppm (LY1).

The quantifications of the main phenolic compounds identified in olive pulp are reported in Table 4. The data highlighted the different rates of hydrolysis and/or solubilization of oleuropein between the treatments only at the beginning of the fermentation, where the concentrations of oleuropein ranged between 1367.09 ppm to 4709.37 ppm, for LY2 and LP respectively. At the end of fermentation, oleuropein in olives ranged between 290 ppm (LY2) and 545 ppm (MIX 1) without significant differences among the treatments.

At the beginning of the trial, hydroxytyrosol in olives contents were between 1991.57 ppm (MIX1) and 3574.43 ppm (WA). During fermentation, it showed an increase only between the first and the second sampling for all treatments, except for SP, which showed constant values until the end of the experiment (~2000 ppm). After the third sampling time, there was a linear decrease in the hydroxytyrosol levels for all treatments except for MIX2 and LY2, which showed a higher content in the last sampling with respect to the previous one. On the 188th day, the hydroxytyrosol values ranged between 1015.06 ppm (MIX1) and 2254.53 ppm (LY2).

The quantification of tyrosol in olives showed a decrease in all treatments between the first and the second sampling, with a depletion ranging between 47% (SP) and 81% (MIX2). Except for SP, showing a deep decrease between the second and the third sampling, the other treatments maintained constant values until the 140th day, which showed a slight increase. Moving from the initial values ranging between 719 ppm (SP) and 1187 ppm (MIX2) to values between 244 ppm (LY1) and 310 ppm (MIX1), it has been shown that this polyphenol is able to pass into brine in all treatments with no significant differences among them.

### 3.3. Textural Proprieties

The puncture test, due to the probe morphology, mainly evaluates the texture properties of the fruit peel. Treatment significantly affected firmness and rigidity index. The sample MIX2 was characterized by the highest firmness and rigidity index and the LY1 by the softest and least rigid peel. The other samples showed quite similar peel characteristics (Table 5).

**Table 5.** Puncture test of final products. Different letters indicate significant differences among the treatment ($p \leq 0.05$). Data are average from three replicates ± standard error.

| Treatments | Firmness (g) | Area (g × mm) | Rigidity Index (g/mm) |
|---|---|---|---|
| SP | 37.59 ± 1.01 [a] | 30.73 ± 1.24 [a] | 20.68 ± 0.56 [ab] |
| LP | 36.21 ± 0.91 [ab] | 28.46 ± 1.01 [a] | 21.01 ± 0.71 [ab] |
| WA | 36.05 ± 0.83 [ab] | 29.30 ± 1.03 [a] | 19.36 ± 0.36 [ab] |
| MIX1 | 36.44 ± 0.90 [ab] | 28.26 ± 0.89 [a] | 20.69 ± 0.55 [ab] |
| LY1 | 33.28 ± 0.96 [b] | 26.68 ± 1.16 [a] | 18.90 ± 0.48 [b] |
| MIX2 | 38.23 ± 1.27 [a] | 30.05 ± 1.27 [a] | 21.75 ± 0.63 [a] |
| LY2 | 36.81 ± 1.04 [b] | 28.04 ± 1.14 [a] | 22.00 ± 0.78 [a] |

The TPA test allowed for the evaluation of the texture of the olive pulp. Except for cohesiveness, all the other parameters were significantly influenced by treatment. The overall data showed that LP, WA, MIX1 and LY1 samples differed from the control SP, for their highest value of hardness, gumminess and chewiness, while MIX2 and LY2 samples seemed to be more similar to the control, even though these differences were not always statistically significant (Table 6).

**Table 6.** Texture Profile Analysis (TPA) of final products. Different letters indicate significant differences among the treatments ($p \leq 0.05$). Data are average from three replicates $\pm$ standard error.

| Treatments | Hardness (g) | Springiness (mm) | Cohesiveness | Gumminess (g) | Chewiness (g) |
|---|---|---|---|---|---|
| SP | 286.35 $\pm$ 11.47 [ab] | 0.66 $\pm$ 0.01 [ab] | 0.50 $\pm$ 0.01 [a] | 142.46 $\pm$ 5.45 [abc] | 96.79 $\pm$ 4.79 [bc] |
| LP | 315.20 $\pm$ 17.71 [ab] | 0.67 $\pm$ 0.01 [ab] | 0.50 $\pm$ 0.01 [a] | 155.40 $\pm$ 8.30 [ab] | 126.90 $\pm$ 6.43 [a] |
| WA | 334.29 $\pm$ 12.26 [ab] | 0.66 $\pm$ 0.01 [ab] | 0.48 $\pm$ 0.01 [a] | 158.83 $\pm$ 5.65 [ab] | 107.31 $\pm$ 4.31 [abc] |
| MIX1 | 337.11 $\pm$ 10.55 [a] | 0.69 $\pm$ 0.01 [a] | 0.48 $\pm$ 0.01 [a] | 162.80 $\pm$ 5.14 [a] | 116.23 $\pm$ 3.90 [ab] |
| LY1 | 321.51 $\pm$ 13.74 [ab] | 0.69 $\pm$ 0.01 [a] | 0.48 $\pm$ 0.01 [a] | 155.81 $\pm$ 6.29 [abc] | 115.08 $\pm$ 4.54 [ab] |
| MIX2 | 277.62 $\pm$ 11.61 [b] | 0.61 $\pm$ 0.01 [c] | 0.48 $\pm$ 0.01 [a] | 130.37 $\pm$ 5.39 [c] | 81.52 $\pm$ 4.04 [c] |
| LY2 | 314.41 $\pm$ 23.30 [ab] | 0.65 $\pm$ 0.01 [cb] | 0.47 $\pm$ 0.01 [a] | 134.42 $\pm$ 5.75 [bc] | 101.06 $\pm$ 6.45 [abc] |

The MIX1 sample was characterized by the highest value of hardness, springiness and gumminess. On the other hand, the MIX2 sample showed the lowest values for all parameters.

## 4. Discussion

Yeast growth during the early phases of fermentation showed a high variability and higher cell concentrations (Figure 1B) compared to what was previously reported for spontaneous fermentations of *Leccino* [27], which is probably due to higher fermentation temperature. Less growth was observed on the 40th day in treatments in which there were no yeast inoculations (statistically significant only for LY1); this could be ascribed to an inhibition of the growth of autochthonous yeasts from killer selected strains, whose growth, due to lower adaptability to the brines with respect to autochthonous yeasts, was lower at the start, which is likely due to a longer lag phase. In particular, this was observed in LY1, where *S. cerevisiae* was added just 10 days before the analysis, and to a lesser extent, also in MIX1, where *S. cerevisiae* was added simultaneously with *L. plantarum* 40 days before the analysis. Greater growth in yeasts was observed where *W. anomalus* was added (WA, MIX2 and LY2); this could denote a major adaptation ability of *W. anomalus* to brines, or its lower killer activity in terms of *S. cerevisiae*. Yeast growth continued until the 77th day, then yeasts' cell concentrations decreased. The maximum cell concentration was reached on the 70th day for LY1 and SP, and on the 77th day for LP, WA, MIX1, MIX2 and LY2. Even if the differences observed were not statistically significant, in general, growth was higher for *W. anomalus* when added in sequential inoculations, and for *S. cerevisiae* when added in mixed inoculations with *L. plantarum*. This could indicate a better synergy between *S. cerevisiae* and LABs, while *W. anomalous* growth could benefit from slowing of the LABs' activity. In the SP treatment, yeasts dropped rapidly on the 84th day; in the other treatments, the decline was slower. MIX1 and LY2 showed the highest yeast cell concentrations.

One possible advantage of the use of starters cultures of yeasts in olive fermentation is the improvement of lactic acid bacterial growth and lactic acid production [9,13]; this effect was not observed in our assays. LABs developed early and declined after the second month (Figure 1A). The addition of starter cultures did not improve their growth with respect to spontaneous fermentations, as previously described by Lanza et al. [12]; however, a positive effect of *W. anomalus*, inoculated alone or mixed with *L. plantarum*, was observed on the 40th and 63rd days, which is statistically significant in terms of the other inoculation strategies involving yeast starters, but not for spontaneous fermentations and inoculations with the *L. plantarum* selected strain. Bleve et al. [27] reported that LABs were undetectable until the 135th day in spontaneous fermentations of *Leccino*, and they reached $1.4 \times 10^5$ CFU/mL maximum cell concentration on day 180, while in our assay, they reached much higher cell concentrations with mean values of $2\text{--}3 \times 10^7$ CFU/mL from the 70th to the 140th day of fermentation; as for yeasts, this could be due to higher fermentation temperatures.

The growth of total aerobic bacteria (Figure 1C) reduced in the third month of fermentation when inocula of both yeasts and LABs were added, but these differences decreased during late fermentation. On the 40th day, their growth was significantly higher in MIX2, as

was also observed for LAB growth. *Enterobacteriaceae* are considered undesired microorganisms, and their presence in processed foods is considered a signal for possible safety risks. In olive fermentation, they can be present during early phases, but they usually disappear soon after the growth of LABs and yeasts [27]. In our assay, their cell concentrations were higher than expected and rose until the fourth month (Figure 1D). Their growth was not prevented by LABs and/or yeasts inoculation. It could have been favored by pH rising and by the availability of mannitol as selective energy and carbon source [28].

Sugars are the main soluble components in olive tissues. During fermentation, they provide energy and carbon sources to microorganisms producing secondary metabolites correlated to the flavor of the product [11]. Raw olive flesh contains both simple sugars, mainly represented by glucose, fructose and sucrose and polyols, like mannitol and sorbitol [29]. Significant differences in sugars and polyols concentrations occur among cultivars, and they are likely related to the genotype and to different climatic and environmental conditions [11].

In our assays, sugars were rapidly consumed in the brines (Figure 2) and their concentration decreased rapidly in olives (Table 2). Among sugar alcohols, mannitol in olives on the 23rd day was unchanged with respect to the raw olives. In the brines, on the 16th day, it was present at concentrations higher than the glucose and fructose, indicating poor consumption from microorganisms. Mannitol is scarcely soluble in water: only 18% *w/v* at 25 °C, then low amounts can dissolve from olives to brines. It was still present on the 40th day in olives and on the 105th day in the brines. In olives on the 105th day, when mannitol disappeared, a significant presence of ethanol was detected (Table 3). Mannitol can be used as an energy and carbon source by several microorganisms, including microorganisms usually found in olives and brines, such as several *Enterobacteriaceae* (i.e., *Klebsiella*, *Serratia*, *Proteus*, *Escherichia coli*) [30], homofermentative LABs (while heterofermentative LABs can produce, but not consume, mannitol), yeasts (including *Saccharomyces*), fungi (including *Fusarium* spp.), and other bacteria (including *Staphylococcus aureus*) [31,32]. The simultaneous disappearance of mannitol and rise of ethanol observed on the 105th day (Table 3) could be ascribed to the development of mannitol-consuming microorganisms inside the olives.

In our assays, mannitol was about 25% of the total sugars and sugar alcohols in the olives, the most abundant after glucose (65%) (Table 3). The growth of *Enterobacteriaceae* (Figure 1D), and the presence of abundant molds, mainly *F. solani*, could be ascribed to high mannitol content, as these microorganisms can use mannitol as a carbon source. *Enterobateriaceae* usually grow early when fermentation starts, then rapidly decrease when the yeasts and LAB become dominant [9], while in our assays they were higher than usually reported for olive fermentations. In the olive tree, mannitol plays a major role in osmoregulation, and it is accumulated as a response to drought stress [33]; therefore, increased drought stress due to climate change could induce an increase in mannitol in olives in the coming years, and consequently, it could modify the equilibrium among microbial populations during fermentation in favor of mannitol-catabolizing microorganisms. Significant attention should be paid to these factors, as an increased risk of potentially pathogenic microorganisms, such as *S. aureus* or *E. coli*, or other spoilage microorganisms, such as molds [34], could arise in olives fermentation. Another sugar alcohol, sorbitol, was also present, but only in traces in raw olives (Table 1), while it rose during fermentations in olives (Table 3) and reached high concentrations in the brines only at late fermentation (Figure 2D). It is known that several yeasts can produce sorbitol, including yeasts that usually grow during olive fermentation, such as *Candida boidinii* and *S. cerevisiae*, whereas LABs do not produce sorbitol, even if some of them can catabolize it (i.e., *L. plantarum*) [31]. Therefore, sorbitol increase in the brines at late fermentation could likely be ascribed to its accumulation in yeast cells as a form of protection from osmotic stress, followed by release during cell lysis. In addition, sorbitol present in the olives could be released in the brines during late fermentation due to lysis induced by hydrolytic enzymes produced, i.e., by molds. Glycerol in olives followed the same trend as sorbitol, but it was not found in the brines (Table 3). Textural property

deterioration (Tables 5 and 6) that indicates the destructuring of cell walls and membranes might result in the decompartmentalization of cellular enzymes, the release of products of degradation (i.e., glycerol from triacylglycerol hydrolysis, glucose from hydrolysis of residual oleuropein), and absorption of compounds from brine (Tables 2–4).

Organic acids in the brines originate from the olives and from microbial metabolism. Citric, malic and succinic acids are typically found as constituents in olives [35]. Citrate can also be produced by molds [36]: this could explain its progressive increase (Figure 4A). Succinic acid as well can be produced by microorganisms usually found in olive fermentation, such as yeasts, *Enterobacteriaceae* and *Propionibacteriaceae* [9,37,38]: it showed a highly variable, bell-shaped curve, which is likely due to the interplay between metabolisms of different microbial populations (Figure 4C). The rapid disappearance of malic acid (Figure 4B) can be expected, as it is easily metabolized by LABs. Lactic, acetic and pyruvic acids, along with ethanol, are the main products of sugars fermentation from heterofermentative LABs. Lactic, acetic and pyruvic acids were found early at high concentrations on the 16th day, then a decrease was observed (Figure 3). The decrease in lactic acid is a significant problem in olive fermentations due to its role in maintaining a low pH. In our assays, on the 140th day, the concentrations of lactic acid, and organic acids in general, were very low, concurrently with a very high pH (Figure 5). This could be ascribed to the appearance of molds from the 63rd day, when the pH rise started, even if also propionibacteria can oxidize lactic acid to acetic acid.

Both lactic and acetic acid played an important role in lowering pH: on the 105th day, the highest pH was found in LP, where the lactic acid concentration was the lowest, and the lowest pH was found in MIX2, where the lactic acid concentration was the highest, and acetic acid was lowest in SP and WA, where pH is high, even if less than in LP. MIX2, where the pH was the lowest, showed the highest concentrations of lactic and acetic acids. Even if the pH gradually and considerably raised until the end of fermentation, its rise was reduced in treatments where both yeasts and LABs were inoculated. The high pH, probably due to mold growth, might have favored the growth of *Enterobacteriaceae*.

The total absence of *Fusarium* spp. on MIX1, the mold responsible for olive softening as *Penicillium* spp. and *Aspergillus niger* [39], may explain the highest values of hardness and gumminess found on MIX1 treatment. Molds and *Enterobacteriaceae* could be responsible for fermented olives softening in all treatments except in MIX1, as shown by the textural proprieties measured by TPA. This result is likely due to the presence of *L. plantarum* and *S. cerevisiae* as inoculum.

These microorganisms could also be responsible for very high ethanol concentrations in the brines (Figure 3D). Ethanol was also found in olives on the 105th day (Table 3), corresponding to the highest development of molds and *Enterobacteriaceae* (Figure 1D). In addition, the slight glucose increase detected in the brines at the end of the assay, difficult to be explained, could be ascribed to cell lytic processes.

The main final purpose of fermentation is to debitter the olives; as expected in the final product, a deep decrease in oleuropein content occurred both in the brines and in the olive pulp. The detected values of oleuropein aglycone, hydroxytyrosol and tyrosol in raw olives were similar to the data reported by Servilli et al. [40] for cv. *Leccino* in terms of oleuropein, while the values of hydroxytyrosol and tyrosol were quite higher than data showed in the same paper. Ranalli et al. [41] recorded similar levels of oleuropein for *Leccino* olives at the same ripening stage (1.33 g/kg of olives). In olive pulp, the different treatments seemed not to influence the final level of oleuropein; in the brine samples, at the end of the experiment, SP and MIX2 showed the highest content of oleuropein. The lowest value of oleuropein was reached in MIX1, so it can be assumed that the use of *S. cerevisiae* had a positive effect on the oleuropein hydrolysis. This result is in good accordance with other papers, where starter-driven fermentations gave the brines with lower content of oleuropein in comparison with spontaneous fermentation [18,42,43]. As hypothesized in several works, in co-inoculated fermentation, a synergic effect occurs between the action of LABs and yeasts [44,45].

In terms of hydroxytyrosol and tyrosol, scarce data are available in the literature for raw drupes of the tested cultivar. As reported by Lanza et al. [4], the hydroxytyrosol content in debittered olives pulp is the consequence of an equilibrium between the solubilization of this molecule from pulp to brine and the enzymatic hydrolysis of oleuropein carried out by LABs. Despite a decrease in hydroxytyrosol during fermentation, the residual quantity of this molecule, ranging between 1000 ppm and 2254 ppm, provides significant nutraceutical potential for the final product and correlates to its antioxidant and antiradical activity. This property is due to the electron-donating ability of the hydroxyl groups in the ortho position, and the subsequent formation of stable intramolecular hydrogen bonds with the phenoxylic radical [46]. In other studies [47], a correlation between the use of olive mill wastewater extracts (rich in hydroxytyrosol) and reduced activities related to tumor cell behavior in lung cancer cell lines was found. The treatment LY2 showed the highest content of hydroxytyrosol, highlighting a synergic effect of LABs and *W. anomalus* inocula in the retention of this precious molecule.

## 5. Conclusions

Starter cultures of selected strains of killer yeasts and LABs assessed as a sustainable way to control olive fermentation by avoiding pretreatments and the use of excessive salt in the brines provide unexpected results. Even if the fermentations started regularly, reaching rapidly low pH and a good level of debittering, an excessive growth of undesirable microorganisms was observed, such as *Enterobacteriaceae* and molds, which induced progressive pH growth, olives softening, and anomalous release in the brines of the olive compounds.

The addition of microbial starters partially reduced the pH rise, in particular when both LABs and yeasts were inoculated, but after six months of fermentation, the brines reached almost neutral pH. The excessive growth of undesirable microorganisms could have been induced by high winter temperatures and the availability of selective carbon and energy sources such as mannitol, whose concentration in olives can be increased by tree drought stress during cultivation. The possible role of climate change on the quality and safety of fermented foods should be investigated further and garner more attention. A possible positive effect on the nutraceutical value of olives due to yeasts and LAB inocula can be signaled with significant production of hydroxytyrosol. Due to the recent attribution of neuroprotective activity to hydroxytyrosol and its several potential therapeutic effects against degenerative and cardiovascular diseases, preserving this compound in the final product is important. The higher presence of this molecule in starter-driven fermented olives suggests that an enhancement of the nutraceutical value of this product can be obtained with further studies involving different combinations of strains of yeasts and LABs.

**Supplementary Materials:** The following supporting information can be downloaded at: https://www.mdpi.com/article/10.3390/fermentation9020182/s1, Figure S1: Morphological description of *Fusarium solani*. (A) Arrow indicates long monophialides; (B) Aerial mycelium presentation: arrow indicates false heads (microconidia in situ); (C) chlamydospores produced in PDA; (D) Macroconidia from sporodochia. Scale bars: 25 μm.

**Author Contributions:** Conceptualization, L.B.; Formal analysis, G.F.B., C.A.M., C.M., L.B., G.C. and M.V.; Investigation, G.F.B., C.M., C.A.M. and M.V.; Supervision, L.B.; Visualization, G.F.B. and C.M.; Writing—original draft, L.B., G.C. and C.A.M.; Writing—review and editing, G.F.B. and C.M. All authors have read and agreed to the published version of the manuscript.

**Funding:** This research was funded by the Italian Ministry of Agricultural, Food, and Forestry Policies (MiPAAF), D.M. 93882/2017, D.M. n. 30654/2019 and D.M. 35902/2019, Project DEAOLIVA, CUP C54I17000020006. The APC was offered by MDPI.

**Institutional Review Board Statement:** Not applicable.

**Informed Consent Statement:** Not applicable.

**Data Availability Statement:** The data presented in this study are available in this article.

**Acknowledgments:** The authors acknowledge Ilaria Mannazzu for her kind willingness to provide yeast strains and operational and conceptual support; Roberto Lo Scalzo for his support in chemical analysis; Alessia Cerullo for her assistance with the formal analysis during her curricular internship, and our colleague Mauro Solomita, in honor of his well-deserved retirement.

**Conflicts of Interest:** The authors declare no conflict of interest.

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
