# Peer review of "Assessment of Starters of Lactic Acid Bacteria and Killer Yeasts: Selected Strains in Lab-Scale Fermentations of Table Olives (Olea europaea L.) cv. Leccino"

_fermentation, doi:10.3390/fermentation9020182_

Round 1

Reviewer 1 Report

The selection of strains of microorganisms that can be used as a starter for the fermentation of olives is an important task. The use of specially selected strains reduces the fermentation time, prevents the growth of undesirable microorganisms and spoilage of the final product.

In this work, the authors used two strains of killer yeast Wickerhamomyces anomalus and Saccharomyces cerevisiae and a strain of Lactobacillus plantarum bacteria to ferment olives and evaluated their effect on pH, soluble sugars, alcohols, organic acids, phenolic compounds, rheological properties of olives compared with spontaneous fermentation. They showed a multidirectional effect of microbial starter cultures, depending on specific microorganisms and the method of inoculation, on the concentration of the studied compounds in olive pulp and brine. Unfortunately, none of the starter variants used prevented the growth of undesirable microorganisms - Enterobacteriaceae. The authors suggest that this may be due to the increased content of mannitol in olives, the increase in the concentration of which may be due to changes in climatic conditions during the growth of olive trees.

The results obtained may stimulate further research on the role of different microbial starters in olive fermentation.

Some questions and comments:

1.It is necessary to give the composition of MRS broth (is Man Rogosa and Sharpe the same?), YEPD, 3M Petrifilm, PDA media or give references to them.

2. Why among the olive fermentation options there is no option using Saccharomyces cerevisiae similar to option ”iii”?

3. Figure 1, 2, 4, 6: Very fine font to indicate significant differences in variants; fine font to indicate fermentation variants. The font should be enlarged! It may be necessary to change the scale of the figures so that the information given in the figures and in the text can be compared.

4. Why are there no initial values of the studied parameters in the tables at the beginning of fermentation?

5. Why is there an increase in the concentration of sucrose, glucose in pulp of Leccino olives on the 140th day compared to the 105th day? (Table 2).

6. The same question regarding the change in the concentration of mannitol, sorbitol and glycerol (Table 3).

7. General note to tables 2-4: It is better to replace " evolution of concentration " with a change in concentration.

7. Is there any explanation for the very different dynamics of changes in the concentration of citric, malic, succinic acids, which are intermediate metabolites of the tricarboxylic acid cycle (Figure 4)?

8. What can explain that "the growth was higher in W. anomalous when added in sequential inoculations, and in S. cerevisiae when added in mixed inoculation with L. plantarum"? (Discussion, lines 442-444).

9. Lines 558-559: If “MIX2 showed the highest content of oleuropein” and “The lowest value of oleuropein was reached in MIX1”, so it can be assumed that the use of lactic acid bacteria and S. cerevisiae had a positive effect on the oleuropein hydrolysis”.

But! Lactic acid bacteria are part of both MIX2 and MIX1, they differ only in the type of yeast. Therefore, it is more correct to talk about the role of S. cerevisiae here.

10. Lines 576-577: “The treatment LY2 showed the highest content in hydroxytyrosol, highlighting a synergic effect of the LABs and yeasts inocula in the retention of this precious molecule.”

It is required to clarify what kind of yeast!

Author Response

Author’s reply to review report (Reviewer 1)

The authors are thankful for the reviewers´ comments because they allow to significantly improve the quality of their work.

1.It is necessary to give the composition of MRS broth (is Man Rogosa and Sharpe the same?), YEPD, 3M Petrifilm, PDA media or give references to them.

We added the compositions and description of all the media and commercial preparations in Section 2.4

  1. Why among the olive fermentation options there is no option using Saccharomyces cerevisiae similar to option ”iii”?

We are aware that a correct experimental set up should have included both Saccharomyces and Wickerhamomyces treatments; however, due to limits in number of jars and space, and mostly in number of samples that could be processed at each sampling time for microbiological analysis, we had to choose among excluding treatments with yeasts strains alone, or with yeasts + LABs. Finally, we considered technologically more significant the association of yeast + LABs. However, as we could find place and time to add a treatment with just one yeast strain, we decided to carry out the treatment (iii) too; then, as the results reached were interesting, we decided to show them, but we can exclude them if Reviewer 1 would consider it necessary.

  1. Figure 1, 2, 4, 6: Very fine font to indicate significant differences in variants; fine font to indicate fermentation variants. The font should be enlarged! It may be necessary to change the scale of the figures so that the information given in the figures and in the text can be compared.

The figures and the fonts were enlarged. The same scale was used in figures when this was possible without losing the possibility to evidence trends and differences among treatments. The enlargement of the font should make the axis legends more readable: this should support a better comprehension of data shown.

  1. 4. Why are there no initial values of the studied parameters in the tables at the beginning of fermentation?

The results of the analysis for the characterization of row olives were reported in Table 1. We considered these data as the starting point of fermentations.

  1. Why is there an increase in the concentration of sucrose, glucose in pulp of Leccino olives on the 140th day compared to the 105th day? (Table 2).
  2. The same question regarding the change in the concentration of mannitol, sorbitol and glycerol (Table 3).

These are very good questions. These data are really amazing, so our first reaction was to interpret the results of 105th day as an analytical error or artifact. But it is also amazing that the same error was present in all treatments at the same sampling time, and only at 105th day; moreover, we found interesting that at the 105th day a considerable growth of molds and Enterobacteriaceae was also observed, that could have accelerated olives destructuring and degradation: affected olives texture could indicate destructuring of cell walls and membranes, decompartmentalization of cellular enzymes, then release of products of degradation (i.e. glucose from hydrolysis of residual oleuropein, glycerol from triacylglycerol hydrolysis), and re-adsorpion of compounds from brines (i.e. mannitol and sorbitol, synthesized by microorganisms as osmoprotectants due to brines high salt concentrations). All these possible explanations are speculations; we proposed them in sentences added at line 568-572

  1. 7. General note to tables 2-4: It is better to replace " evolution of concentration " with a change in concentration.

Done

  1. Is there any explanation for the very different dynamics of changes in the concentration of citric, malic, succinic acids, which are intermediate metabolites of the tricarboxylic acid cycle (Figure 4)?

Due to the very complex interplay between the metabolisms of different microbial populations and also of olive cells, it is very difficult to explain the dynamics of each organic acid. For example, citric, malic and succinic acids are constituents of olives drupes as well as intermediates in TCA, and they can be metabolized by microorganisms. We have divided in two different figures the organic acids coming only from microbial metabolism (Figure 3) and organic acids that are constituents of olives drupes, but that can be also catabolized or produced by microorganisms. The decrease of malic acid can be explained by the prevailing activity of lactic acid bacteria that can convert it in lactic acid; citric acid increase could be explained by its synthesis by molds, and succinic acid probably follows the generic trend of TCA intermediate. We reformulated these statements at lines 573-583

  1. What can explain that "the growth was higher in W. anomalous when added in sequential inoculations, and in S. cerevisiae when added in mixed inoculation with L. plantarum"? (Discussion, lines 442-444).

A possible explanation is stated in the sentence added at lines 496-7: “This could indicate a better synergy between S. cerevisiae and LABs, while W. anomalous could benefit for its growth from slowing down of LABs activity.”

  1. Lines 558-559: If “MIX2 showed the highest content of oleuropein” and “The lowest value of oleuropein was reached in MIX1”, so it can be assumed that the use of lactic acid bacteria and S. cerevisiae had a positive effect on the oleuropein hydrolysis”.

But! Lactic acid bacteria are part of both MIX2 and MIX1, they differ only in the type of yeast. Therefore, it is more correct to talk about the role of S. cerevisiae here.

The sentence was rephrased as follows: “The lowest value of oleuropein was reached in MIX1, so it can be assumed that the use of S. cerevisiae had a positive effect on the oleuropein hydrolysis” (Lines 626-7).

  1. Lines 576-577: “The treatment LY2 showed the highest content in hydroxytyrosol, highlighting a synergic effect of the LABs and yeasts inocula in the retention of this precious molecule.”

It is required to clarify what kind of yeast!

The sentence was rephrased as follows: “The treatment LY2 showed the highest content in hydroxytyrosol, highlighting a synergic effect of LABs and W. anomalous inocula in the retention of this precious molecule.” (Line 644)

Reviewer 2 Report

Manuscript entitled “Assessment of starters of Lactic Acid Bacteria and Killer Yeasts  selected strains in lab-scale fermentations of table olives (Olea 3 europaea L.) cv. Leccino” by Bencresciuto et al., is about the evaluation of the impact of different starters combinations on table olives fermentation. The assessment of the growth dynamics of the different microbial groups revealed the unexpected growth of undesirable microorganisms, with possible safety risk, likely due to both a marked pH increase during fermentation and the availability of fermentable carbon sources, such as mannitol whose concentration in olives may increase following tree draught stress.  

The manuscript is clearly organized and well written. Moreover, it highlights the need to further investigate on the impact of climate change on quality and safety of fermented olives.

There are some concerns regarding the graphical representation of the results. I would suggest the authors to enlarge the size of the font utilized in figures (very difficult to read, numbers and letters) and possibly to utilize the same scale in the ordinate axis where possible, in order to facilitate the comparison of the effect of different starters and inoculation strategies on the parameters analysed.

Other questions regard the anomalous pH increase during table olives fermentation. Have other authors reported such a problem during table olives fermntation? Was brine pH evaluated at inoculations?

Author Response

Author’s reply to review report (Reviewer 2)

The authors are thankful for the reviewers´ comments because they allow to significantly improve the quality of their work.

There are some concerns regarding the graphical representation of the results. I would suggest the authors to enlarge the size of the font utilized in figures (very difficult to read, numbers and letters) and possibly to utilize the same scale in the ordinate axis where possible, in order to facilitate the comparison of the effect of different starters and inoculation strategies on the parameters analysed.

The figures and the fonts were enlarged. The same scale was used in ordinate axis when this was possible without losing the possibility to evidence trends and differences among treatments. The enlargement of the font should make the axis legends more readable: this should support a better comprehension of data shown.

Other questions regard the anomalous pH increase during table olives fermentation. Have other authors reported such a problem during table olives fermntation? Was brine pH evaluated at inoculations?

We only found a 6.2 pH mentioned (as unshown data) about a fermentation with mixed (bacterium + yeast) inoculum (L. plantarum B1 + Candida boidinii) in:

Lanza, B.; Di Marco, S.; Bacceli, M.; Di Serio, M.G.; Di Loreto, G.; Cellini, M.; Simone, N. Lactiplantibacillus plantarum Used as Single, Multiple, and Mixed Starter Combined with Candida boidinii for Table Olive Fermentations: Chemical, Textural, and Sensorial Characterization of Final Products. Fermentation 2021, 7, 239. https://doi.org/10.3390/fermentation7040239

We evaluated the brine pH after sterilization and before inoculation: it was 6.5.
